# Reconfigurable structured light generation in a multicore fibre amplifier

Di Lin [1✉], Joel Carpenter [2], Yutong Feng [1], Saurabh Jain[1], Yongmin Jung[1], Yujun Feng[1], Michalis N. Zervas[1] & David J. Richardson[1]

Structured light, with spatially varying phase or polarization distributions, has given rise to many novel applications in fields ranging from optical communication to laser-based material processing. However the efficient and flexible generation of such beams from a compact laser source at practical output powers still remains a great challenge. Here we describe an approach capable of addressing this need based on the coherent combination of multiple tailored Gaussian beams emitted from a multicore fibre (MCF) amplifier. We report a proof-of-concept structured light generation experiment, using a cladding-pumped 7-core MCF amplifier as an integrated parallel amplifier array and a spatial light modulator (SLM) to actively control the amplitude, polarization and phase of the signal light input to each fibre core. We report the successful generation of various structured light beams including high-order linearly polarized spatial fibre modes, cylindrical vector (CV) beams and helical phase front optical vortex (OV) beams.

[1] Optoelectronics Research Centre, University of Southampton, Southampton SO17 1BJ, UK. [2] School of Information Technology and Electrical Engineering, The University of Queensland, Brisbane, QLD 4072, Australia. ✉email: di.lin@soton.ac.uk

Structured light with a reconfigurable spatial beam distribution in terms of amplitude, phase and polarization has emerged as an attractive tool for a variety of photonics applications due to the great control it provides for light-matter interactions. In particular, OV beams with a helically varying phase front carrying orbital angular momentum (OAM) are already used for a wide range of applications, such as macro-/nano-particle manipulation[1,2], microscopy[3,4], quantum information[5] and both free space and fibre optic communications[6,7]. Moreover CV beams that exhibit a spatially varying polarization (e.g. radial or azimuthal polarization) have found applications in laser-based materials processing[8,9], particle acceleration[10], optical trapping[11] and high-resolution imaging[12]. Many approaches to generate such spatially structured beams have been developed that either rely upon reshaping the optical field external to the laser cavity, or directly generating structured light from a suitably designed laser cavity. For example, OV beams can be generated by employing external beam-shaping optics such as a spiral phase plate[13], or a liquid crystal SLM[14], and CV beams can be obtained by using spatially inhomogeneous birefringent optics[15,16], or by controllably interfering two orthogonally polarized beams[17,18]. Among these, the digitally controllable SLM-based approach offers a great deal of flexibility and in principle allows the generation of any desired spatial mode. However, it suffers from a relatively low efficiency and a very limited power handling capability (typically < 2 W cm$^{-2}$)[18]. On the other hand, the direct generation of such beams from laser cavities/amplifiers offers advantage of higher mode purity and much higher power scalability. The underlying mechanism in this instance is to provide higher net gain for the specific mode relative to all others by using specially designed mode selective bulk or fibre-based optical elements within the laser cavity/amplifier[19–29].

A few methods that provide some flexibility in mode selection have been implemented in solid-state lasers but generally this has been achieved at the cost of a high intra-cavity loss and the output powers have been limited by damage to the special optics used due to the high intra-cavity powers involved[23,24]. Alternatively, adaptive spatial mode shaping has been achieved in large mode area fibre amplifiers by employing fibre-based photonic lanterns and deformable mirrors as beam shaping elements for high power operation, enabling efficient generation of scalar linearly polarized (LP) modes[30–34]. Similarly, coherent beam combination (CBC) using signals from multiple (independent) parallel laser sources has recently been proposed as a promising way to form exotic beams that avoids the low damage threshold problem of such flexible and scalable beam shaping elements and has been investigated theoretically and experimentally[35–41]. In this approach, the active beam control element(s) can be positioned before the amplifier(s) (where the power levels are relatively modest) to provide the control needed for active phase locking between the multiple output beams at the (higher power) system output. Such master oscillator power amplifier (MOPA) based configurations might provide significant scope for power scaling of beam-shaped fibre lasers[41]. In addition, CBC from MCF amplifier/lasers has previously been successfully demonstrated as an effective way for power scaling of simple Gaussian beams[42–44].

In this paper we experimentally demonstrate the generation of various forms of complex beam (including azimuthally phase-chirped OV beams and radially/azimuthally polarized CV beams defined on the higher-order Poincaré sphere (HOPS)) by coherent combination of individually amplified signals in a MCF amplifier. Use of the MCF architecture provides for a highly integrated array of amplifiers and drastically reduces system complexity by enabling sharing of components across the multiple spatial channels, e.g. isolators, pump diodes and the SLM that we used to control the amplitude, phase and polarization

state of the signal light launched into each of the individual cores. Furthermore, since all the cores experience similar thermal and mechanical perturbations due to their very similar and close physical proximity, the environmental sensitivity of the relative phase and polarization between signals propagating through the different individual cores is substantially reduced compared to the case of arrays of physically separate single core fibre amplifiers. This leads to a corresponding reduction in active beam control requirements.

## Results

**Working principle.** Consider an MCF with $N$ uncoupled single-mode cores that are evenly arranged on a circle. The cores are sufficiently separated to ensure negligible inter-core crosstalk and each core behaves as an independent waveguide such that the total electric field at the fibre output can be described (under a Gaussian mode approximation) as the sum of the individual beams:

$$\mathbf{E}(x,y,z=0) = \sum_{j=1}^{N} \mathbf{a}_j \cdot \exp\left(-\frac{(x-x_j)^2 + (y-y_j)^2}{w_0^2}\right) \quad (1)$$

Where $\mathbf{a}_j = a_{xj}\exp(i\varphi_{xj})\hat{\mathbf{x}} + a_{yj}\exp(i\varphi_{yj})\hat{\mathbf{y}}$ is the complex amplitude of the fundamental Gaussian beam of the $j$-th core, in which $\hat{\mathbf{x}}, \hat{\mathbf{y}}$ are unit vectors, $a_{x(y)j}$ and $\varphi_{x(y)j}$ are the amplitude and phase of the $x(y)$-component of Gaussian beam of the $j$-th core, respectively; $w_0$ is the effective mode field radius of each core, and $(x_j, y_j)$ is the center coordinate of the $j$-th core. Although the output beam of each core can be considered an independent beamlet, the far-field of the MCF would typically be complex speckle pattern due to the interference between the individual beamlets with essentially random amplitude and phases. However, by properly tailoring the relative amplitude, phase and polarization state of the individual beamlets, various complex electric fields can be effectively synthesized in the far-field. Figure 1 shows some simulation examples of several different spatial modes generated from the coherent combination of 6 beamlets in a tiled-aperture configuration with particular phase and polarization distributions. One of most familiar is that of a linearly polarized Gaussian-shaped beam arising from in-phase coherent superposition of all beamlets and a petal-like beam by

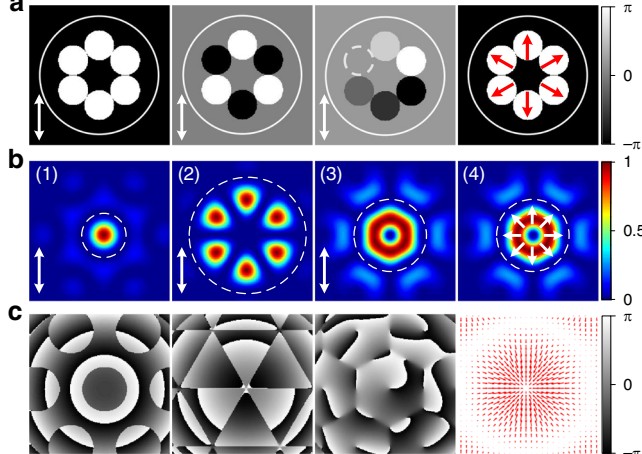

**Fig. 1 Calculation of coherently combined modes from a 6-core MCF. a** Examples of phase and polarization distributions of the MCF beamlets for the generation of various structured light beams. **b, c** Calculated far-field (**b**) intensity, (**c**) phase or polarization state distributions of the synthesized modes (white and red arrows show the polarization orientations).

the exactly out-of-phase (i.e. $\pi$ phase shift) superposition between adjacent beamlets, as shown in the first two columns. Additionally, more complex beams can be formed through more sophisticated control of the individual beamlets. For instance, the coherent superposition of $N$ beamlets can result in a beam with azimuthal phase variation (phase term $\exp(il\varphi)$, where $\varphi$ is the azimuthal angle) when the relative phase of the $j$-th beamlet is set to be $2\pi l(j-1)/N$ under the same polarization state. The optical beam will carry OAM of $l\hbar$ per photon ($l < N/2$) and the third column of Fig. 1 shows the first order ($l = 1$) OAM beam generated from an $N = 6$ combination. In addition, a CV beam can be realized by controlling the polarization states of the individual beamlets and the fourth column of Fig. 1 shows an example of a radially polarized beam with spatially varying polarization. Here, each beamlet has a linear polarization which rotates by an angle of $2\pi l(j-1)/6$ from its original orientation. It is worth mentioning that the beam combining efficiency (defined as the ratio of the power contained within the encircled white dash line to the total power in the beam, as in Fig. 1b) varies for different structured light beams. Our numerical calculation (see Supplementary Note 1) shows that the combination efficiencies of the individual beams shown in Fig. 1b(1–4) are ~46%, ~90%, ~60% and ~60%, respectively. The higher combining efficiency of the higher-order modes can be attributed to the greater similarity of the intensity distribution of these combined beams to the input beamlet array, resulting in a lesser power contribution to the side lobes.

**Experimental set-up**. A schematic of our experimental setup is shown in Fig. 2 (described in more detail in the Methods section). A single input laser is used to seed the individual cores of a cladding-pumped Er:Yb-doped MCF amplifier operating at a wavelength of 1550 nm via a beam-shaper based on a phase-only SLM[45]. An Er:Yb-doped fibre embodiment was chosen in the first instance due primarily to fibre and component availability in our laboratory, however in principle the technique should be applicable to other fibre dopants operating in other spectral regimes (e.g. ytterbium for laser wavelengths around 1060 nm and thulium for wavelengths around 2000 nm). The beam-shaper enables independent control of the amplitude, phase and polarization state of the seed light launched into each of the individual MCF cores. The output of the MCF amplifier is coherently combined in the far-field using a tiled-aperture configuration. The far-field output beam is then characterized using an appropriate spatial mode correlation filter and the resulting correlation signal is then used in an iterative optimisation process to adjust the SLM control parameters in order to achieve the desired beam-combined modal profile at the system output.

**Coherent beam shaping procedure**. Although the individual cores of the (uncoupled) MCF can be considered to be independent waveguides with almost identical optical path lengths, small refractive index variations between the individual cores (e.g. due to residual birefringence, fibre fabrication imperfections, or fibre macro-/micro-bending) result in a difference in the effective propagation constants/birefringence. This then results in significant differences in the phase and polarization states of the output signals amplified in different cores. Without any input signal control, beam combination in the far-field from a MCF amplifier tends to result in a speckle intensity pattern with a random polarization state. However, adaptive input wavefront shaping using a suitable optimisation algorithm provides an effective way to generate an amplified structured beam at the output of a MCF amplifier. In the instance of a Gaussian target output beam then transmission through an on-axis pinhole can

be used to define a merit function for the iterative optimisation process. However, a more complex matched spatial filter is required in the case of higher-order spatial modes, due to the more complex phase and polarization distributions involved. In this instance, to provide a robust and unambiguous merit function, we construct a suitable spatial mode correlation-filter to compare the output beam pattern with the target output mode[46,47]. It is well known that when a collimated output beam is incident on a correlation filter with a transmission function $T(r) = \Psi_{\mathrm{des}}^{*}(r)$ (where $\Psi_{\mathrm{des}}^{*}(r)$ denotes the complex conjugate of the desired mode) the on-axis intensity in the far-field (back focal plane of a lens) is proportional to the power in the output beam within that desired mode. The on-axis signal power can be detected by coupling the light into a SMF positioned at the back focal plane. The launched power into the SMF can then be measured using a power meter and used as a merit function to provide a feedback signal to the beam-shaper to set the desired amplitude, phase and polarization of the combined output beam via an iterative optimisation process. We use a standard steepest gradient descent algorithm (which is widely used for solving high dimensional optimisation problems) to optimise the individual beamlet parameters including the amplitude and phase in each polarization state resulting in a total of $4 \times N$ ($N$ is the number of cores) parameters to be optimised. For the generation of beams resembling the scalar LP fibre modes we used correlation-filters based on the relevant phase-plate and a LPO and for the generation of complex HOPS modes we used a QP in combination with a QWP and a LPO.

**Phase and polarization control**. In a preliminary experiment, we launched horizontally linearly polarized seed laser light into each core of the MCF without any phase control (i.e. with a uniform initial phase and polarization front at the MCF input). As shown in Fig. 3a, the combined output beam profile in the far-field exhibited significant distortion and is far from the Gaussian beam anticipated for ideal in-phase locking. The insets in Fig. 3a shows the output beam profiles on the two orthogonal polarization axes after passing the beam through a LPO that indicate that the original linear polarization is not preserved after amplification within the MCF amplifier. The Stokes parameters of the output beam polarization were also measured (using the method described in ref. [48]) and the combined output beam in this instance was found to be partially polarized with a polarization purity of ~63%. The blue line in Fig. 3c shows the polarization ellipse of the polarized component of the output beam without any input control (i.e. the beam shown in Fig. 3a). It is clear that the launched linearly polarized input beam has evolved into an elliptically polarized beam after amplification through the MCF. These observations indicate that not only the relative phase but also the polarization state of the individual beamlets needs to be controlled on a per-core basis for effective CBC in a MCF amplifier.

We first tested the feasibility of our beam-shaping concept in a conventional in-phase CBC process targeting a linearly polarized Gaussian beam. In this case, the correlation-filter was made of a LPO with its transmission axis aligned to the horizontal direction. After iterative optimisation, the initial distorted output beam (Fig. 3a) was gradually converted into a beam with a Gaussian-shaped intensity profile as shown in Fig. 3b. The insets in Fig. 3b show the intensity profiles of the two orthogonal polarization components and almost null intensity was observed in the vertical direction, indicating that the output beam is highly linearly polarized in the horizontal direction. The degree of linear polarization was quantified by measuring the Stokes parameters and the resulting polarization ellipse is plotted in Fig. 3c (red

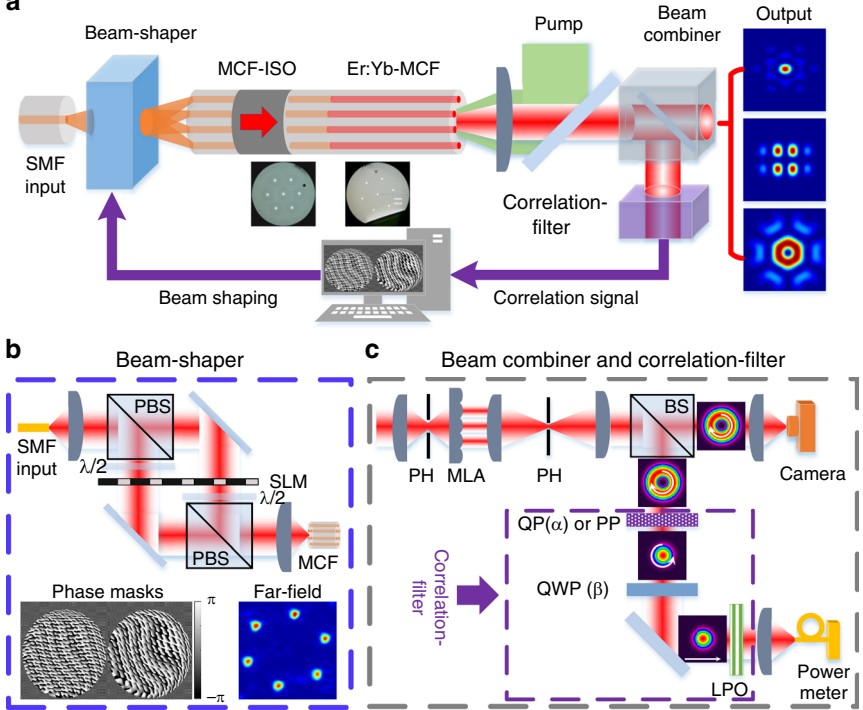

**Fig. 2 Schematic of the experimental setup. a** MCF amplifier: a single seed laser beam is adaptively shaped by a spatial beam-shaper and the resultant field used to excite the individual cores of the MCF amplifier; multiple amplified beams are coherently combined to obtain various spatial modes in the far-field. **b** Beam-shaper: the input beam is split into the two orthogonal linear polarizations, each of which is split into N beams: the phases and amplitudes of these two components are then independently modulated by the SLM. The $N$ pairs of orthogonally linear-polarized beams are recombined to form $N$ beams with well-defined polarizations and phases, and are then coupled into the individual cores of the MCF. An example of phase masks displayed on the SLM and the generated 6 independent beam spots in the plane of the MCF core. **c** Adaptive coherent beam combination setup: the multiple output beamlets are collimated by a microlens array (MLA) and are combined by a focusing lens in the far-field; the shaped beam is characterized by a spatial mode correlation-filter consisting of a properly aligned $q$-plate (QP) (or a phase plate (PP)), a quarter-waveplate (QWP) and a linear polarizer (LPO). The resulting correlation signal is detected by a single-mode fibre (SMF) pigtailed power meter, where the measured power is used as the merit function for the iterative optimisation algorithm. ISO isolator; PBS polarization beam splitter; PH pin-hole; BS polarization-insensitive beam splitter.

line). The polarization extinction ratio (PER) was measured to be ~15 dB and the measured output power variation with polarizer rotation angle is well described by the expected cosine squared angular dependency as shown in Fig. 3d.

**Scalar LP mode generation**. We next examined the generation of various higher order scalar LP fibre modes by adjusting the input wavefront to the MCF amplifier. Here, we placed the relevant phase-plate of the desired LP mode in front of the LPO to construct the relevant correlation-filter. A few theoretical examples of the complex amplitudes for the individual beamlets required to generate specific LP modes are illustrated in Fig. 4, along with theoretical and experimental far field beam profiles. The experimentally measured far-field beam profiles agree well with the corresponding theoretical predictions indicating that the complex amplitude of the individual beamlets is controlled as required. The two-dimensional correlation coefficients of the measured intensities with respect to the theretical predicitons were calculated to be >~0.97 for the $LP_{01}$, $LP_{11}$ and $LP_{21}$ modes, and ~0.93 for the $LP_{31}$ mode (see Supplementary Note 2 for more details). The slightly lower correlation coefficient of the $LP_{31}$ mode is likely due to residual aberrations in the beam combining system. The polarization state of the output beam was also linear in the horizontal direction and the measured PER was >13 dB. It is worth mentioning that not all cores need to be excited to generate specific target modes depending on the symmetry, e.g. only 4 cores were excited in order to generate the $LP_{11}$ and the $LP_{21}$

modes, whereas 6 cores were excited for the $LP_{01}$ and the $LP_{31}$ modes.

**Generation of HOPS beams**. Finally, we investigated the generation of advanced HOPS ($|l| = 1$) beams. Each point on the HOPS represents a doughnut-shaped beam with a unique polarization state as shown in Fig. 5a, b. The points on the equator represent spatially varying linearly polarized beams including a radial or azimuthal polarization and the poles represent right- and left-handed circularly polarized OV beams with helical phase front of opposite handedness. Due to the spatially varying phase and polarization states of the HOPS beams, the critical issue is to construct an appropriate spatial correlation-filter in the beam diagnosis setup to obtain an unambiguous correlation signal for the desired mode. It is well known that a linearly polarized Gaussian-shaped beam can be converted to any unique HOPS beam by passing it through a QWP and a QP set with a proper rotation angle of $\alpha$ and $\beta$, respectively[24,49] (see "Methods" for more details). We use the reverse process in our correlation-filter i.e. we look to convert the HOPS beams to a linearly polarized Gaussian-shaped beam.

As a preparatory step before generating the HOPS beams, the phase and polarization states of the six outer cores of our MCF were first controlled to create a linearly polarized Gaussian-shaped output beam. After confirming this, the QP was then inserted and aligned to be concentric with the output Gaussian beam as shown in Fig. 2c. The concentricity can be experimentally confirmed by the observation of a clean doughnut-shaped

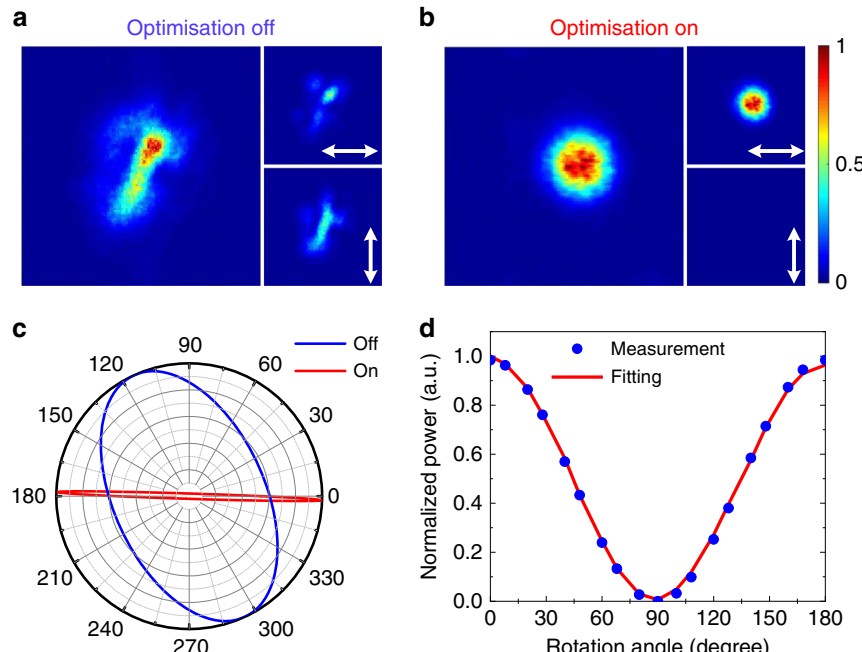

**Fig. 3 Experimental demonstrations of depolarization and phase mismatch in a MCF amplifier and the requirement for active input control to generate a linearly polarized Gaussian beam. a** When optimisation is off (without any wavefront and polarization shaping for the beamlet input signals), the measured far-field intensity and the two orthogonal polarization components after passing the combined beam through a linear polarizer (white arrows show the transmission axes of the linear polarizer). Significant depolarization and phase mismatch were observed after amplification through the MCF amplifier. **b** After optimisation of the wavefront and polarization state of the input signal (i.e. Optimisation ON), the measured far-field intensity and the horizontal and vertical polarization components of the output beam. In this case, a well-defined linearly polarized Gaussian beam was obtained. c, The measured polarization ellipses for the output beams of **a**, **b** showing that the initial elliptical polarization was efficiently converted to the desired linear polarization under active control. **d** The output power measured by rotating a linear polarizer in the case of active control. The angular dependency is again well described by a cosine squared fit and the measured PER was ~15 dB.

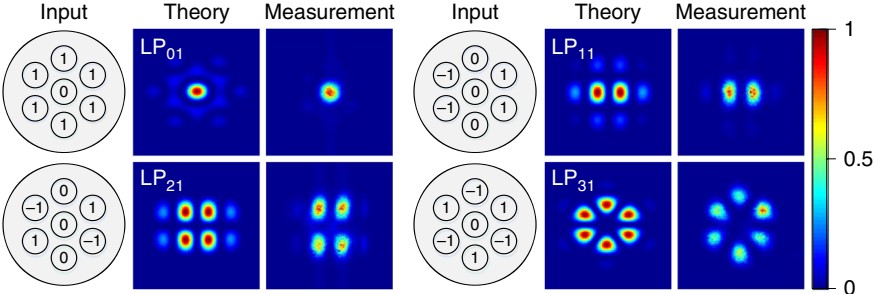

**Fig. 4 Scalar LP mode generation.** Idealised complex amplitudes of the individual cores required to generate specific LP fibre modes and the corresponding beam profiles in the far-field (in theory and measurement).

beam on a CCD camera located in the far-field in conjunction with a corresponding drastic reduction in the power launched into the SMF (typically < 20 dB). We generated several structured beams represented on the equator of the HOPS by setting the QWP with $\beta = 0°$ and rotating the QP ($\alpha$). For $\alpha = 0°$, a doughnut-shaped beam was obtained through our iterative optimisation process as shown in Fig. 5c. Note that it took ~10–15 min to convert the Gaussian-shaped beam to the desired doughnut-shaped beam and an exemplary plot of the normalized merit value as a function of optimisation time is presented in Supplementary Fig. 3. The beam was then passed through a rotatable linear polarizer to examine the spatial polarization distribution. As shown in Fig. 5d, we observed a clear two lobed intensity pattern and the orientation of the two lobes was parallel to the transmission axis of the linear polarizer, confirming that the output beam is a radially polarized vector beam. With $\alpha$

rotated by 90°, a similar doughnut-shaped beam was obtained but an azimuthally polarized vector beam was created as shown in Fig. 5e, f. In this case, orientation of the two lobes is perpendicular to the transmission axis of the linear polarizer. Given that the centre core of the MCF was unseeded, strong amplified stimulated emissions (ASE) in both the forward and backward directions is to be expected as the pump power is increased. In order to avoid damage to the MCF isolator due to back-propagating ASE we limited the gain of our MCF amplifier to be ~20 dB obtaining a total output power of ~530 mW (measured directly after our MCF amplifier) in both cases. The amplified multiple beams were then combined with a MLA and a focusing lens and the combined beam was measured to be ~230 mW after a PH in Fig. 2c, corresponding to a combination efficiency of ~44%, which is ~16% lower than the theoretical value of ~60%. The difference is probably due to the optical

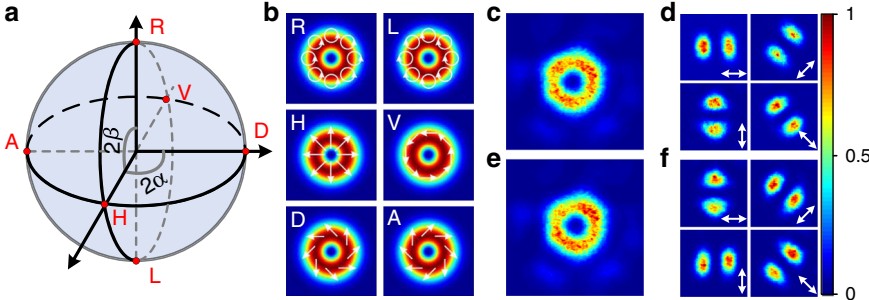

**Fig. 5 HOPS beams. a** HOPS for the order of |*l*| = 1. The points on the equator represent spatially variant linearly polarized beams. Particularly, the points H and V represent the radially and azimuthally polarized beam, respectively. The points D and A corresponds to spiral linear polarization states. The poles R and L represent right-handed and left-handed circularly polarized optical vortex beams with opposite handedness of helical phase-front. **b** Theoretical beam intensity and polarization distributions for the six particular points in **a** (white arrows indicate the electric field polarization direction). **c** Experimentally measured radially polarized output beam profile and **d**, two lobed structure intensity profiles when the beam in **c** passed through a rotated linear polarizer. The two lobes' directions are parallel to the transmission axis of the linear polarizer verifying the radial polarization. **e**, **f** Experimentally measured azimuthally polarized beam profile and its polarization state, where the two lobes' direction is perpendicular to the orientation of the linear polarizer.

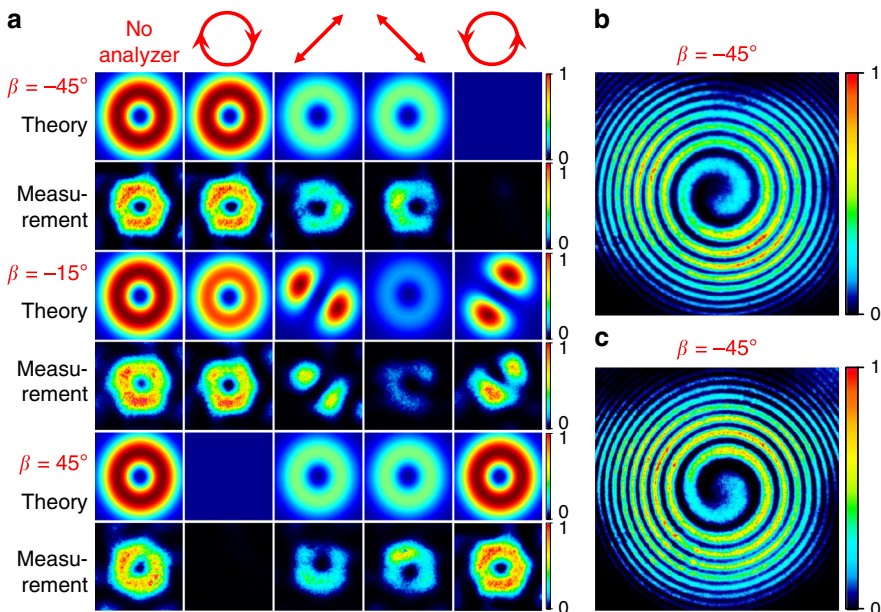

**Fig. 6 OAM beams. a** Theoretical (odd rows) and experimentally measured (even rows) output beams and the corresponding intensity profiles after passing through a rotated circular polarizer in the case of *β* = ±45° and *β* = −15°. **b** The measured spiral interference fringes of the generated vortex beams with opposite handedness when *β* = ±45°.

aberrations in the beam combination system induced by slight misalignment of the composite optical elements. For instance any deviation of the beam from the optical axis and inappropriately collimated beamlets can drastically reduce the combination efficiency (see more details in Supplementary Note 4). The modal purity was measured to be ∼93% according to the vector mode decomposition method described in ref. [47], indicating that the relative phase and polarization states of each core are well controlled.

We have also tested other points on the HOPS as shown in Fig. 5a In this case, the matching correlation-filter was adjusted by rotating the QWP from *β* = −45° to *β* = 45° with a fixed angle of *α* = 0° for the QP, resulting in an evolution of the output beam from a left circularly polarized OV mode to a right circularly polarized OV mode with opposite handedness of helical phase-front. Figure 6a shows three examples of output beam profiles measured in our experiment together with the spatial intensity patterns after passage through a rotated circular polarizer. It can be seen that the measured intensity profiles (even rows) are well

matched to the theoretical predictions (odd rows) and a near null intensity was observed in one of the circular polarizations when *β* = ±45°. It indicates that a clean circularly polarized OV beam was obtained with a high polarization purity. The PER was measured to be ∼12 dB. These circularly polarized beams were further analysed by interfering them with a reference beam with a spherical wavefront and the measured beams were confirmed to be OAM modes with a single spiral fringe and opposite handedness as shown in Fig. 6b.

It is worth mentioning that our coherently combined structured beams are quite stable and repeatable in the laboratory environment. Importantly, the applied SLM phase pattern did not need to be changed once the optimisation process was completed with the generated output beams preserved for periods in excess of 1 h (the longest we left our experiment running on a given mode). Figure 7 shows the normalized power of the correlation signal collected by the SMF as a function of time at the maximum output power of the radially polarized beam in open loop conditions (i.e. with a constant SLM phase pattern). A small

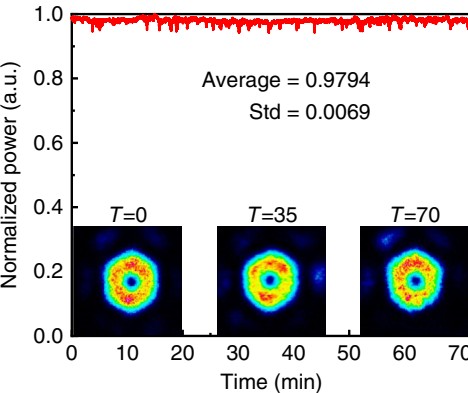

**Fig. 7 Long-term stability.** The normalized power of the correlation signal collected by the SMF, indicating stable phase locking over 70 min. The inserts are examples of beam intensity profiles measured at different times, showing that the doughnut-shaped beam was well preserved during the measurement period.

power fluctuation (~6% peak-to-valley) was observed with an average value of 0.979 and a root mean square of 0.007 over ~70 min, verifying stable polarization and phase locking across six cores in our MCF amplifier. This implies very relaxed bandwidth requirements for CBC systems based on MCF amplifiers should active feedback stabilisation be deemed necessary in a given application. In addition, the optimised SLM phase patterns for various structured light spatial modes could be determined and the modes then generated on-demand simply by applying the corresponding phase pattern.

## Discussion

In order to generate more sophisticated structured light beams with a tailored amplitude, phase and polarization distribution, the 7-core MCF amplifier used in our experiment would need to be further scaled to a higher core-count. Naturally, this would require more advanced MCF design/fabrication with respect to the number of cores and the associated core arrangement. For example, in order to generate an OAM mode with a topological charge of $l$, a minimum number of $3l$ cores should be used in a single circle. With recent advances in MCF technology, MCF amplifiers with 49 cores have been recently demonstrated[50] and these high core count MCF amplifiers not only lead to a high degree of spatial freedom but offer even higher output powers/energies beyond the nonlinear limit of a single-mode fibre amplifier/laser. The use of an uncoupled MCF effectively avoids mode coupling and the laser beam shaping can be simplified and stabilized by the simultaneous control of the complex amplitudes of $N$ independent single-mode fibre lasers. It also significantly reduces the number of optimisation iterations compared with the situation in multimode fibres or coupled-core MCFs with severe mode coupling. In addition, it is fundamentally immune to transverse spatial gain competition and a high mode purity is expected at high gain because the output beam is the result of the coherent superposition of all independent single-mode outputs.

In conclusion, we have proposed and demonstrated an approach to efficiently generate flexibly controlled structured light beams from a MCF amplifier based on the CBC technique. By tailoring the complex amplitude of the input signal to the MCF amplifier, we overcame depolarization and phase mismatch in the MCF amplifier and successfully generated various spatial modes in the far-field, including conventional scalar fibre LP modes and exotic HOPS beams. This work paves the way to the generation of high power and reconfigurable structured light

beams from simple and compact laser sources. The switching and high power structured light beams may offer advantages in a wide range of applications such as laser material processing, free-space optical communication and high-resolution imaging.

## Methods

**Experimental setup.** The experimental setup is illustrated in Fig. 2a. It starts with a pre-amplified laser source at 1550 nm with a SMF output and a clean Gaussian beam is injected into the beam-shaper (Fig. 2b). The beam-shaper is composed of a reflective phase-only liquid crystal SLM (Holoeye PLUTO-2-TELCO-013) in combination with some polarization diversity optics. The SLM has an active area of $15.36 \times 8.64$ mm comprising $1920 \times 1080$ pixels, corresponding to a pixel pitch of 8 μm with a filling factor of 93%. It can operate at any wavelength from 1400 to 1700 nm with a reflectivity of 80% and a damage threshold of ~2 W cm$^{-2}$. The input beam is collimated and divided into two partial beams with orthogonal polarizations using a PBS. A half-waveplate ensures that the polarization orientation of the two beams is aligned to the SLM. The area of the SLM is divided in two halves to display the phase masks which split each of the input beam into $N$ Gaussian beams with controlled phase and amplitude. A half-waveplate and a PBS are used to recombine the reflected $N$ pairs of orthogonal linearly polarized beam to form $N$ beamlets with well-defined polarizations and phases, and a lens with a focal length of 20 mm is used to couple the shaped light into the passive MCF in the output Fourier plane. Although illustrated in a transmissive topology for clarity in Fig. 2b, in practice each beam reflects off its own phase mask displayed on the SLM. The required phase pattern displayed on the SLM is simply the phase of the Fourier transform of the required spots in the plane of the MCF core. This is a super-position of blazed gratings each weighted by the complex amplitude of its corresponding spot in the plane of the MCF core[45]. The insertion loss of our beam-shaper (from the input SMF to the output facet of the passive MCF) was measured to be ~4.1 dB for each core.

As a proof-of-principle experiment, an in-house 7-core Er/Yb-doped MCF was used and a cladding pumped amplifier configuration was implemented to simultaneously amplify multiple spatial beams. As shown in the cross-sectional image of the active MCF in Fig. 2a, seven step-index circular cores are arranged in a hexagon with a core-to-core spacing of 50 μm and each core has a diameter of 5 μm with a numerical aperture (NA) of 0.22. The large core-to-core spacing effectively prevents crosstalk between adjacent cores and the measured fibre crosstalk was less than −50 dB, meaning each core operates independently. The active MCF has an inner cladding diameter of 195 μm and a low-index acrylate polymer coating (~280 μm) to create a double clad structure and to enable efficient pump light coupling from a high power multimode pump laser diode. Each MCF core is co-doped with $Er^{3+}$ (~0.2 wt %) and $Yb^{3+}$ (~1.3 wt %) and the fibre cladding absorption at the pump wavelength of 975 nm was measured to be ~1.25 dB m$^{-1}$ per core. The fibre was ~5.5 m long and was coiled with a diameter of ~15 cm on an aluminium plate. The output end facet of the active MCF was polished with an angle of ~9° and a home-made MCF isolator was spliced at the input of the active MCF to suppress parasitic lasing of each core and to prevent any counter-propagating light damaging the SLM. Light from the SLM was free-space coupled directly into the input MCF isolator pigtail. Our passive MCF was designed/fabricated to provide a good match to the active fibre. It has the same core pitch distance (50 μm) and cladding diameter (~195 μm). However, the core parameters are somewhat different (core diameter of 8 μm with an NA of 0.12). The measured splice loss between the two MCFs was ~0.8 dB due to the mode field diameter mismatch. The passive MCF has a high-index acrylate polymer coating to attenuate any signal light inadvertently coupled into the cladding. The insertion loss of the MCF isolator was measured to be ~1.5–3.9 dB with some variation between cores, which is mainly due to the imperfect coupling in the free-space part of the isolator. However, the signal power launched into each core of the active MCF can be equalized by adjusting the relative power coupled to the passive MCF via the beam-shaper. The multimode pump beam was free-space coupled into the inner cladding of the MCF through a dichroic mirror (reflection at 975 nm and transmission at 1550 nm).

A schematic of the beam combination setup is depicted in Fig. 2c. The output beam was first magnified by a factor of 10 in order to match the lens pitch of an MLA, which results in a beam separation of 500 μm between the neighboring beamlets. The MLA has a focal length of 12 mm in a hexagonal arrangement and it can effectively collimate the individual beamlets with an increased beam filling factor in the near-field, providing the great benefit of combination efficiency improvement in the form of a tiled-aperture configuration. A focusing lens with a focal length of 200 mm was placed behind the MLA to achieve CBC in the far-field with the generation of a variety of structured light beams. A PH was put in the focal plane to block any undesired cladding modes. Afterwards, the output beam was split into two partial beams using a BS. One beam is re-imaged by a CCD camera for monitoring the intensity profiles of the CBC and the other beam was passed through a mode correlation-filter allowing the combined beam to be characterized and to provide feedback to the computer controlled SLM for adaptive adjustment of the complex amplitude of the input signal. The bandwidth of the feedback loop is 4 Hz, determined by the time required to allow the SLM to update the phase mask, measure the correlation signal and associated software based data processing.

**Working principle of the *q*-plate.** Consider a Gaussian-shaped beam with horizontal linear polarization incident on a QWP in sequence with a *q*-plate, the resultant polarization state can be derived according to the Jones matrix as follows[24,49]:

$$\mathbf{E}_{\text{out}} = \cos(\pi/4 + \beta)\exp(-i(\alpha - \beta))|L_l\rangle + \sin(\pi/4 + \beta)\exp(i(\alpha - \beta))|R_l\rangle \quad (2)$$

where $|l| = 2q$ is the topological charge, $\alpha$ and $\beta$ are the rotation angles of the *q*-plate and the QWP, respectively; $|L_l\rangle = e^{-i|l|\varphi}|L\rangle$, $|R_l\rangle = e^{i|l|\varphi}|R\rangle$, where $|L\rangle$ and $|R\rangle$ refer to left and right circularly polarized state, respectively. Eq. (2) describes any polarization state on the HOPS determined by the rotation angle of $\alpha$ and $\beta$. Note that, $\beta$ determines the relative weight of the two basis OAM modes and the relative angle between the *q*-plate and QWP ($\alpha$–$\beta$) determines their relative phase. According to Eq. (2), it can be seen that a pure OAM mode is obtained when $\beta = \pm\pi/4$, represented on the poles in Fig. 5a. When $\beta = 0$, the superposition of two opposite OAM modes with equal weight results in spatially variant linear polarization states represented on the equator of the HOPS. In particular, the point H ($\alpha = 0$) represents the radial polarization and the point V ($\alpha = \pi/2$) represents the azimuthal polarization state. Other points between the poles and equator represent elliptically polarized CV beams.

## Data availability

All source data supporting this study are openly available from the University of Southampton Repository (https://doi.org/10.5258/SOTON/D1442)[51].

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

## Acknowledgements

The work was funded by the UK Engineering and Physical Sciences Research Council (EPSRC) through the projects: EP/P012248/1, EP/P027644/1, EP/P030181/1 and EP/N00762X/1. J.C. is supported from the Discovery (DP170101400, DE180100009) program of the Australian Research Council (ARC). We thank Johan Nilsson for providing the seed laser source and thank Anna Peacock and Li Shen for providing the EDFA.

## Author contributions

D.L. conceived the idea, performed the experiments with assistance from Y.F., S.J. and Y.F., and performed the data analysis; D.L. and Y.F. performed the mathematical analysis; J.C. devised the beam-shaper; Y.J. fabricated the MCF isolator; M.N.Z. and D.J.R. supervised the project; D.L., Y.J. and D.J.R. wrote the paper with inputs from all co-authors.

## Competing interests

The authors declare no competing interests.
