## [Peer Review File · Nature Communications]

REVIEWER COMMENTS

Reviewer #1 (Remarks to the Author):

This paper reports a new method to generate high-power reconfigurable structured light, including arbitrary vortex and/or vector beams. It is based on the multicore-fiber amplification of a set of "beamlets" distributed in a given pattern. The independent control of amplitude, phase and polarization of each beamlet allows one a large freedom in controlling the resulting superposition occurring in the far field. A feedback control is then used to finely adjust these quantities in order to obtain a given target structured field. This proposed method is particularly interesting to achieve structured light of very high power, as other methods to structure an incoming beam of light (such as SLMs or q-plates) are usually not viable for very high powers.

There is no very fundamental innovation in this work, as the approach based on combining a set of beamlets for structuring light was already known and multi-core fiber amplifiers are also well known. But the overall method itself is novel and efficient, it seems also convenient and flexible, and could prove very valuable for certain applications. The paper is also well presented and convincing. Overall, I think this work satisfies the criteria of importance and interest needed to be published in Nature Communications.

Reviewer #2 (Remarks to the Author):

The authors present a study and experimental demonstration of beam shaping using a 7-core fibre (with hexagonal arrangement of the cores) employing tiled aperture coherent combination. In general, this paper is interesting, and the results are of good quality, so I would like to recommend it for publication. The only thing that refrains me from doing so is that the manuscript, in its present form, is marred in some passages by a lack of details and, sometimes, (slight) inaccuracies. These should be addressed and corrected before I can give my approval to the publication of the paper, since the lack of information makes it difficult to judge whether the results are good enough for a journal such as Nature Communications. Let me share with you some of my observations (in the other of appearance in the paper).

In the introduction the authors should be a little bit more exhaustive on their references to other works. For example, there has been some work on CV beams using an all-fiber setup that, in my opinion, should be referenced here. The references are:

*Christoph Jocher, Cesar Jauregui, Christian Voigtländer, Fabian Stutzki, Stefan Nolte, Jens Limpert, and Andreas Tünnermann, "Fiber based polarization filter for radially and azimuthally polarized light," *Opt. Express* 19, 19582-19590 (2011)

*Christoph Jocher, Cesar Jauregui, Martin Becker, Manfred Rothhardt, Jens Limpert and Andreas Tünnermann, "An all-fiber Raman laser for cylindrical vector beam generation," *Laser Phys. Lett.* 10 125108 (2013)

Additionally, there has been some work on adaptive beam shaping using photonic lanterns that should also be cited, since the physical underlying principle is somehow similar to the one used in the paper. This references are:

*Juan Montoya, Chris Aleshire, Christopher Hwang, Nicolas K. Fontaine, Amado Velázquez-Benítez, Dale H. Martz, T.Y. Fan, and Dan Ripin, "Photonic lantern adaptive spatial mode control in LMA fiber amplifiers," *Opt. Express* 24, 3405-3413 (2016)

*G. Lopez-Galmiche, Z. Sanjabi Eznaveh, J. E. Antonio-Lopez, A. M. Velazquez Benitez, J. Rodriguez Asomoza, J. J. Sanchez Mondragon, C. Gonnet, P. Sillard, G. Li, A. Schülzgen, C. M. Okonkwo, and R. Amezcua Correa, "Few-mode erbium-doped fiber amplifier with photonic lantern for pump spatial mode

control," Opt. Lett. 41, 2588-2591 (2016)

*Sergio G. Leon-Saval, Nicolas K. Fontaine, Joel R. Salazar-Gil, Burcu Ercan, Roland Ryf, and Joss Bland-Hawthorn, "Mode-selective photonic lanterns for space-division multiplexing," Opt. Express 22, 1036-1044 (2014)

Even though the authors mention this afterwards, I would like that they say right up front that they use a tiled aperture combining approach in the description of the working principle. This would make the operating principle easier to understand.

It would be nice if the authors could include the theoretical combining efficiency of the usable (central) part of the beams presented in Fig. 1. Just for the reader to get an idea of whether more complex beams incur in higher combining penalties or not...

Please include more information about the SLM: number of pixels, pixel size, aperture, operating wavelength region, etc. Likewise, please describe in the main text in more detail the excitation scheme (some of these details can be found in the long figure captions, but the reader should not be forced to look for the information there). This would improve readability.

Also include more information about the MCF: fiber outer diameter, polymer coating or not, doping concentrations, etc. Additionally, from Fig. 2 it seems that the cores are surrounded by a low-index trench. If this is the case please mention and describe it. Additionally, if the low index trenches are there, why do the authors use such an enormous (in relation to the core diameter) core-to-core distance? Is it really necessary to avoid core-to-core cross-talk?

Which are the insertion losses of the beam-shaper? How high is the coupling efficiency of the shaped beam into the MCF?

What is the bandwidth of the feedback loop?

Why do you have the power unbalance seen among the lobes of the beam shown in Fig.4 (for the LP31)?

Which are the correlation values of the shaped beams with the ideal beam? Please provide these values for all the measurements.

Do you have any problems with self-lasing in the configurations in which you left some cores unseeded? Would it be possible to increase the output power further or those self-lasing (or any other effects) limit this? Please elaborate on these topics in the manuscript.

Please include how long does your system need to reach the optimized output beam.

Even though the authors claim that their measured 44% combination efficiency value is "close" to the expected theoretical value of 61%, I do not agree. I think that both values are very far away. So the authors should tone down their claim, explain why they didn't reach the theoretical expectation and what can be done to improve this result. I am also surprised about the "high expected" combination efficiency. 60% is about the maximum that you can get when generating a Gaussian beam in the far field. Thus, I would have expected that the combination efficiency of more complex beams is somewhat lower. Please, to give readers like me a better feeling of what can be expected in terms of combining efficiency, include the theoretical combining efficiency in all your examples.

The authors claim that their system is essentially operating in open loop. If this is the case then, how sensitive is the situation to changes in temperature and/or the exact position of the fiber? How stable are the optimum solutions in time? When you go to the lab in the morning do you need to look for a new optimum solution each day or does the one from the previous day still work as intended?

The authors mention that conventional CBCs always require an active stabilization and control loop. This is

inaccurate. If you only want to have the temporal stability shown in Fig.7, then you do not need any kind of feedback loop with CBC either. However, if you want to improve on that, then you would need a feedback loop both in conventional CBC and in the system presented by the authors. In essence, I do not think that their system is intrinsically more stable than a conventional CBC. Please justify or remove this sentence.

Provided that all these points are addressed I would have enough information to judge whether the results are good enough for Nature Communications (and in principle they certainly look like they are).

Best regards,

NCOMMS-20-06957, “Reconfigurable structured light generation in a multicore fibre amplifier,” written by Di Lin, Joel Carpenter, Yutong Feng, Saurabh Jain, Yongmin Jung, Yujun Feng, Michalis N. Zervas, David J. Richardson

Reviewer comments appear here:

Reviewer #1 (Remarks to the Author):

This paper reports a new method to generate high-power reconfigurable structured light, including arbitrary vortex and/or vector beams. It is based on the multicore-fiber amplification of a set of “beamlets” distributed in a given pattern. The independent control of amplitude, phase and polarization of each beamlet allows one a large freedom in controlling the resulting superposition occurring in the far field. A feedback control is then used to finely adjust these quantities in order to obtain a given target structured field. This proposed method is particularly interesting to achieve structured light of very high power, as other methods to structure an incoming beam of light (such as SLMs or q-plates) are usually not viable for very high powers.

There is no very fundamental innovation in this work, as the approach based on combining a set of beamlets for structuring light was already known and multi-core fiber amplifiers are also well known. But the overall method itself is novel and efficient, it seems also convenient and flexible, and could prove very valuable for certain applications. The paper is also well presented and convincing. Overall, I think this work satisfies the criteria of importance and interest needed to be published in Nature Communications.

Response: We would like to thank the reviewer for this positive evaluation. As the reviewer mentioned, the idea of coherent combination of multiple beamlets with active phase control has been proposed previously [35-41] and a few experimental demonstrations have been reported in physically separated single mode fibre amplifier arrays [36, 41]. In this paper, we further explore this idea to tailor both the phase and polarization states of multiple beamlets to achieve more complex structured light beams such as optical vortex beams and cylindrical vector beams. More importantly, we have employed a multicore fibre architecture for a highly integrated array of amplifiers and drastically reduced system complexity by sharing of components across the multiple spatial channels. This leads to a corresponding reduction in active beam control requirements and provides a promising way to generate flexible high power structured laser beams at source.

Reviewer #2 (Remarks to the Author):

The authors present a study and experimental demonstration of beam shaping using a 7-core fibre (with hexagonal arrangement of the cores) employing tiled aperture coherent combination. In general, this paper is interesting, and the results are of good quality, so I would like to recommend it for publication. The only thing that refrains me from doing so is that the manuscript, in its present form, is marred in some passages by a lack of details and, sometimes, (slight) inaccuracies. These should be addressed and corrected before I can give my approval to the publication of the paper, since the lack of information makes it difficult to judge whether the results are good enough for a journal such as Nature Communications. Let me share with you some of my observations (in the other of appearance in the paper).

Response: We greatly appreciate the reviewer’s detailed and constructive comments. We have carefully considered all of the comments and tried to respond to all significant issues raised by the reviewer on a point-by-point basis as detailed below.

Comment 1: In the introduction the authors should be a little bit more exhaustive on their references to other works. For example, there has been some work on CV beams using an all-fiber setup that, in my opinion, should be referenced here. The references are:

*Christoph Jocher, Cesar Jauregui, Christian Voigtländer, Fabian Stutzki, Stefan Nolte, Jens Limpert, and Andreas Tünnermann, "Fiber based polarization filter for radially and azimuthally polarized light," Opt. Express 19, 19582-19590 (2011)

*Christoph Jocher, Cesar Jauregui, Martin Becker, Manfred Rothhardt, Jens Limpert and Andreas Tünnermann, "An all-fiber Raman laser for cylindrical vector beam generation," Laser Phys. Lett. 10 125108 (2013)

Additionally, there has been some work on adaptive beam shaping using photonic lanterns that should also be cited, since the physical underlying principle is somehow similar to the one used in the paper. This references are:

*Juan Montoya, Chris Aleshire, Christopher Hwang, Nicolas K. Fontaine, Amado Velázquez-Benítez, Dale H. Martz, T.Y. Fan, and Dan Ripin, "Photonic lantern adaptive spatial mode control in LMA fiber amplifiers," Opt. Express 24, 3405-3413 (2016)

*G. Lopez-Galmiche, Z. Sanjabi Eznaveh, J. E. Antonio-Lopez, A. M. Velazquez Benitez, J. Rodriguez Asomoza, J. J. Sanchez Mondragon, C. Gonnet, P. Sillard, G. Li, A. Schülzgen, C. M. Okonkwo, and R. Amezcua Correa, "Few-mode erbium-doped fiber amplifier with photonic lantern for pump spatial mode control," Opt. Lett. 41, 2588-2591 (2016)

*Sergio G. Leon-Saval, Nicolas K. Fontaine, Joel R. Salazar-Gil, Burcu Ercan, Roland Ryf, and Joss Bland-Hawthorn, "Mode-selective photonic lanterns for space-division multiplexing," Opt. Express 22, 1036-1044 (2014)

Response: Thanks for recommending several good and relevant papers. We have added the first two references recommended by the reviewer in the first paragraph of the introduction along with 3 new additional references of our own choice. The other three papers recommended by the reviewer and two new papers of our own choice have been included in the first paragraph on page 2 to provide information on adaptive beam shaping in large mode area fibre amplifiers by employing either a photonic lantern or a deformable mirror as beam shaping elements.

Modification on Page 1, Paragraph 1:

"On the other hand, the direct generation of such beams from laser cavities/**amplifiers** offers advantage of higher mode purity and much higher power scalability. The underlying mechanism in this instance is to provide higher net gain for the specific mode relative to all others by using specially designed mode selective **bulky or fibrized** optical elements within the laser cavity/**amplifier** [19-29]."

Modification on Page 2, Paragraph 1:

Alternatively, adaptive spatial mode shaping has been achieved in large mode area fiber amplifiers by employing fibre-based photonic lanterns and deformable mirrors as beam shaping elements for high power operation, enabling efficient generation of scalar linearly-polarized (LP) modes [30-34]. Similarly, coherent beam combination (CBC) using signals from multiple (independent) parallel laser sources has recently been proposed as a promising way to form exotic beams that avoids the low damage threshold problem of such flexible and scalable beam shaping elements and has been investigated theoretically and experimentally [35-41]."

New references:

[25] C. Jocher, C. Jauregui, C. Voigtlander, F. Stutzki, S. Nolte, J. Limpert, and A. Tünnermann, "Fiber based polarization filter for radially and azimuthally polarized light," Optics Express 19, 19582-19590 (2011).

[26] C. Jocher, C. Jauregui, M. Becker, M. Rothhardt, J. Limpert, and A. Tünnermann, "An all-fiber Raman laser for cylindrical vector beam generation," Laser Physics Letters 10, 125108 (2013).

[27] R. S. Chen, J. H. Wang, X. Q. Zhang, A. T. Wang, H. Ming, F. Li, D. Chung, and Q. W. Zhan, "High efficiency all-fiber cylindrical vector beam laser using a long-period fiber grating," Optics Letters 43, 755-758 (2018).

[28] D. Lin, N. Baktash, S. U. Alam, and D. J. Richardson, "106 W, picosecond Yb-doped fiber MOPA system with a radially polarized output beam," Optics Letters 43, 4957-4960 (2018).

- [29] W. D. Zhang, L. G. Huang, K. Y. Wei, P. Li, B. Q. Jiang, D. Mao, F. Gao, T. Mei, G. Q. Zhang, and J. L. Zhao, "Cylindrical vector beam generation in fiber with mode selectivity and wavelength tunability over broadband by acoustic flexural wave," *Optics Express* **24**(2016).
- [30] S. G. Leon-Saval, N. K. Fontaine, J. R. Salazar-Gil, B. Ercan, R. Ryf, and J. Bland-Hawthorn, "Mode-selective photonic lanterns for spacedivision multiplexing," *Optics Express* **22**, 1036-1044 (2014).
- [31] J. Montoya, C. Aleshire, C. Hwang, N. K. Fontaine, A. Velazquez-Benitez, D. H. Martz, T. Y. Fan, and D. Ripin, "Photonic lantern adaptive spatial mode control in LMA fiber amplifiers," *Optics Express* **24**, 3405-3413 (2016).
- [32] G. Lopez-Galmiche, Z. S. Eznaveh, J. E. Antonio-Lopez, A. M. V. Benitez, J. R. Asomoza, J. J. S. Mondragon, C. Gonnet, P. Sillard, G. Li, A. Schulzgen, C. M. Okonkwo, and R. A. Correa, "Few-mode erbium-doped fiber amplifier with photonic lantern for pump spatial mode control," *Optics Letters* **41**, 2588-2591 (2016).
- [33] R. Florentin, V. Kermene, J. Benoist, A. Desfarges-Berthelemot, D. Pagnoux, A. Barthelemy, and J. P. Huignard, "Shaping the light amplified in a multimode fiber," *Light Sci. Appl* **6**(2017).
- [34] N. Wang, J. C. A. Zacarias, J. E. Antonio-Lopez, Z. S. Eznaveh, C. Gonnet, P. Sillard, S. Leon-Saval, A. Schulzgen, G. F. Li, and R. Amezcua-Correa, "Transverse mode-switchable fiber laser based on a photonic lantern," *Opt. Express* **26**, 32777-32787 (2018).

Comment 2: Even though the authors mention this afterwards, I would like that they say right up front that they use a tiled aperture combining approach in the description of the working principle. This would make the operating principle easier to understand.

Response: We fully agree with the reviewer and a new description has been added in paragraph 3 on page 2:

Modification on Page 2, Paragraph 3:

"Figure 1 shows some simulation examples of several different spatial modes generated from the coherent combination of 6 beamlets in a tiled-aperture configuration with particular phase and polarization distributions."

Comment 3: It would be nice if the authors could include the theoretical combining efficiency of the usable (central) part of the beams presented in Fig. 1. Just for the reader to get an idea of whether more complex beams incur in higher combining penalties or not...

Response: As suggested, we have undertaken work to numerically calculate the combining efficiency for various structured light beams investigated in our experiment. The fundamental Gaussian beam (Figure 1.b(1)) has a calculated combining efficiency of ~46%, the doughnut-shaped optical vortex (Figure 1.b(3)) and cylindrical vector beam (Figure 1.b(4)) both have a calculated combining efficiency of ~60%, and the petal-like beam (Figure 1.b(2)) has a calculated combining efficiency of ~90%. Our results show that the higher-order modes have higher combining efficiency than the fundamental Gaussian beam, which can be attributed to the greater similarity between intensity distribution of these combined beams to the original beamlet array. To clarify this, a new sentence has been added on page 3 and the detailed calculation procedure we adopted is presented in the Supplementary note 1.

Modification on Page 3, Paragraph 1:

"It is worth mentioning that the beam combining efficiency (defined as the ratio of the power contained within the encircled white dash line to the total power in the beam, as in **Figure 1.b**) varies for different structured light beams. Our numerical calculation (see Supplementary note 1) shows that the combining efficiencies of the individual beams shown in **Figure 1.b(1-4)** are ~46%, ~90%, ~60% and ~60%, respectively. The higher combining efficiency of the higher-order modes can be attributed to the greater similarity of the intensity distribution of these combined beams to the input beamlet array, resulting in a lesser power contribution to the side lobes."

Comment 4: Please include more information about the SLM: number of pixels, pixel size, aperture, operating wavelength region, etc. Likewise, please describe in the main text in more detail the excitation scheme (some of these details can be found in the long figure captions, but the reader should not be forced to look for the information there). This would improve readability.

Response: As suggested, more detailed information on the SLM and the beam excitation scheme are provided in the Methods section on page 10.

Modification on Page 10, Paragraph 4:

“The beam-shaper is composed of a reflective phase-only liquid crystal SLM (Holoeye PLUTO-2-TELCO-013) in combination with some polarization diversity optics. The SLM has an active area of 15.36×8.64 mm comprising 1920×1080 pixels, corresponding to a pixel pitch of 8 μm with a filling factor of 93 %. It can operate at any wavelength from 1400 nm to 1700 nm with a reflectivity of 80 % and a damage threshold of ~2 W/cm². The input beam is collimated and divided into two partial beams with orthogonal polarizations using a PBS. A half-waveplate ensures that the polarization orientation of the two beams is aligned to the SLM. The area of the SLM is divided in to two halves to display the phase masks which split each of the input Gaussian beam into N Gaussian beams with controlled phase and amplitude. A half-waveplate and a PBS are used to recombine the reflected N pairs of orthogonal linearly-polarized beam to form N beamlets with well-defined polarizations and phases, and a lens with a focal length of 20 mm is used to couple the shaped light into the passive MCF in the output Fourier plane.”

Comment 5: Also include more information about the MCF: fiber outer diameter, polymer coating or not, doping concentrations, etc. Additionally, from Fig. 2 it seems that the cores are surrounded by a low-index trench. If this is the case please mention and describe it. Additionally, if the low index trenches are there, why do the authors use such an enormous (in relation to the core diameter) core-to-core distance? Is it really necessary to avoid core-to-core cross-talk?

Response: All cores in both the passive and active MCFs have simple step-index refractive index profiles without any trench layer. Although it looks like that there are low index trench structures (i.e. a faint circular boundary around each core) in our passive MCF (Figure 2a) this simply originates from the glass stacking boundary. It results from the use of glass with very slightly different properties in our preform fabrication process. We have now included further detailed specifications of the MCFs in the second paragraph on page 11.

Modification on Page 11, Paragraph 2:

“The active MCF has an inner cladding diameter of 195 μm and a low-index acrylate polymer coating (~280 μm) to create a double clad structure and to enable efficient pump light coupling from a high power multimode pump laser diode. Each MCF core is co-doped with Er³⁺ (~0.2 wt%) and Yb³⁺ (~1.3 wt%) and the fibre cladding absorption at the pump wavelength of 975 nm was measured to be ~1.25 dB/m per core. The fibre was ~5.5 m long and was coiled with a diameter of ~15 cm on an aluminium plate. The output end facet of the active MCF was polished with an angle of ~9 degree and a home-made MCF isolator was spliced at the input of the active MCF to suppress parasitic lasing of each core and to prevent any counter-propagating light damaging the SLM. Light from the SLM was free-space coupled directly into the input MCF isolator pigtail. Our passive MCF was designed/fabricated to provide a good match to the active fibre. It has the same core pitch distance (50 μm) and cladding diameter (~195 μm). However, the core parameters are somewhat different (core diameter of 8 μm with an NA of 0.12). The measured splice loss between the two MCFs was ~0.8 dB due to the mode field diameter mismatch. The passive MCF has a high-index acrylate polymer coating to attenuate any signal light inadvertently coupled into the cladding.”

Comment 6: Which are the insertion losses of the beam-shaper? How high is the coupling efficiency of the shaped beam into the MCF? What is the bandwidth of the feedback loop?

Response: The insertion loss of the beam-shaper was measured to be ~4.1dB, which was measured from the input single-mode fibre to our passive MCF. The loss mainly originates from the SLM (a reflectivity of ~80%, diffraction efficiency of ~80%, and optical loss induced by surface aberrations) and the other optical elements used in the setup. There are two other losses in our MCF amplifier that need to be taken into account: the insertion loss of the MCF isolator and the splice loss between the passive and active MCF (~0.8 dB). The MCF isolator was fabricated in house using micro-optic collimators and the insertion loss was measured to be ~1.5-3.9dB (some variation between cores). The variation is mainly due to the imperfect coupling conditions in the free-space part of the isolator. However, this variation can be balanced by adjusting the relative power coupled to the passive MCF in the beam shaper.

The bandwidth of the feedback loop is determined by the time required to allow the SLM to update the phase mask, power measurement and data communication between the power meter and software. The total time interval was ~250 ms, corresponding to a bandwidth of 4Hz for the feedback loop. The relatively low phase drift due to the inherent common path propagation in the MCF amplifier (as reported in Refs. [42-43]) results in very modest (< 1 Hz) loop bandwidth requirements and is one of the key advantages of using MCF based amplifiers for CBC systems.

In order to address the insertion loss of the beam shaper and the bandwidth of the feedback loop we have added a few descriptions on page 11 and page 12.

Modification on Page 11, Paragraph 1:

“The insertion loss of our beam-shaper (from the input SMF to the output facet of the passive MCF) was measured to be ~4.1dB for each core.”

Modification on Page 11, Paragraph 2:

“The insertion loss of the MCF isolator was measured to be ~1.5-3.9dB with some variation between cores which is mainly due to the imperfect coupling conditions in the free-space part of the isolator. However, the signal power launched into each core of the active MCF can be equalized by adjusting the relative power coupled to the passive MCF via the beam-shaper.”

Modification on Page 11, Paragraph 3:

“The bandwidth of the feedback loop is 4Hz, determined by the time required to allow the SLM to update the phase mask, measure the correlation signal and associated software based data processing.”

Comment 7: Why do you have the power unbalance seen among the lobes of the beam shown in Fig. 4 (for the LP₃₁)?

Response: The intensity imbalance for the LP₃₁ mode in Figure 4 is simply associated with the modal purity. The two-dimensional correlation coefficient of the measured intensity with respect to the theoretical predictions is calculated to be >0.97 for the LP₀₁, LP₁₁ and LP₂₁ modes but ~0.93 for the LP₃₁ mode (see Supplementary note 2 for more details). Due to the reduced modal quality of the LP₃₁ mode, we can observe some level of imbalance in the intensity distribution due to the mode beating with other spatial modes. The slightly lower correlation coefficient of the LP₃₁ mode is likely caused by residual aberrations in the beam combining system. Figure R 1 (b) shows one example of the calculated coherently combined LP₃₁ mode with unbalanced intensity distribution due to the small lateral displacement of the MLA with respect to the optical axis ($\Delta x = 2.5\mu\text{m}$, $\Delta y = 2.5\mu\text{m}$, 0.5% of the microlens pitch).

Modification on Page 7, Paragraph 2:

“The two-dimensional correlation coefficients of the measured intensities with respect to the theoretical predictions were calculated to be > 0.97 for the LP₀₁, LP₁₁ and LP₂₁ modes, and ~0.93 for the LP₃₁ mode (see Supplementary note 2 for more details). The slightly lower correlation coefficient of the LP₃₁ mode is likely due to residual aberrations in the beam combining system.”

Figure R 1. (a) The intensity profile of the measured LP_{31} mode; (b) the calculated intensity profile of the coherently combined LP_{31} mode when the MLA has a slight lateral displacement with respect to the optical axis ($\Delta x = 2.5\mu\text{m}$, $\Delta y = 2.5\mu\text{m}$, 0.5% of the microlens pitch).

Comment 8: Which are the correlation values of the shaped beams with the ideal beam? Please provide these values for all the measurements.

Response: Please refer to our response to Comment #7.

Comment 9: Do you have any problems with self-lasing in the configurations in which you left some cores unseeded? Would it be possible to increase the output power further or those self-lasing (or any other effects) limit this? Please elaborate on these topics in the manuscript.

Response: In our experiment, the output end facet of the active MCF was angle-polished and a home-made MCF isolator was spliced at the input of the active MCF and we did not observe any self-lasing in the unseeded cores. However, the amplified stimulated emission (ASE) in the back propagating direction in the unseeded core increased linearly with the increasing pump power and this would ultimately cause damage to the MCF isolator if the pump power were increased too far. Note that the MCF isolator used in our experiments was originally designed and fabricated for optical communications experiments for which high power operation was not required. In principle, the output power of our amplifier could be further increased by applying more pump power or by adding additional amplifier stages, but this is currently limited by the power handling capacity of the MCF isolator used. Further power scaling of the output beam will be carried out in future work by fabricating high power optical isolators and by using a more efficient MCF gain medium (e.g. based on Yb^{3+} or Tm^{3+}) with optimized core/cladding arrangements.

Modification on Page 9, Paragraph 1:

“Given that the centre core of the MCF was unseeded, strong amplified stimulated emissions (ASE) in both the forward and backward directions is to be expected as the pump power is increased. In order to avoid damage to the MCF isolator due to back-propagating ASE we limited the gain of our MCF amplifier to ~20 dB obtaining a total output power of ~530 mW (measured directly after our MCF amplifier) in both cases.”

Comment 10: Please include how long does your system need to reach the optimized output beam.

Response: It takes about ~10-15 minutes to reach the optimized output beam. Figure R 2 plots one example of the merit value as a function of optimisation time when converting the Gaussian-shaped beam into a doughnut-shaped CV beam in the experiment.

Modification on Page 8, Paragraph 2:

“Note that it took about 10-15 minutes to convert the Gaussian-shaped beam to the desired doughnut-shaped beam and an exemplary plot of the normalized merit value as a function of optimization time is presented in Figure S3 in Supplementary note 3.”

Figure R 2 Normalized merit value vs optimisation time.

Comment 11: Even though the authors claim that their measured 44% combination efficiency value is “close” to the expected theoretical value of 61%, I do not agree. I think that both values are very far away. So the authors should tone down their claim, explain why they didn’t reach the theoretical expectation and what can be done to improve this result. I am also surprised about the “high expected” combination efficiency. 60% is about the maximum that you can get when generating a Gaussian beam in the far field. Thus, I would have expected that the combination efficiency of more complex beams is somewhat lower. Please, to give readers like me a better feeling of what can be expected in terms of combining efficiency, include the theoretical combining efficiency in all your examples.

Response: We appreciate the suggestion and rephrased the sentence in Page 9 from “~ corresponding to a combination efficiency of ~44 %”. This value is close to the theoretical value of ~60 %” to “~ corresponding to a combination efficiency of ~44% which is ~16 % lower than the theoretical value of ~60 %”. The difference is probably due to optical aberrations in the beam combination system induced by slight misalignment of the composite optical elements. For instance any deviation of the beam from the optical axis and inappropriately collimated beamlets can drastically reduce the combination efficiency (see more details in Supplementary note 4). Figure R 3(a) shows the calculated beam combination efficiency of the doughnut-shaped OV beams when the position of the beam waist of the magnified beamlets deviates from the front focal plane of the MLA (z is the distance between the beam waist and the MLA). It can be seen that defocus by ~15% ($\Delta z = 1.8\text{mm}$, given that the focal length of the MLA is 12mm) can reduce the combination efficiency below 50%. Figure R 3(b) shows the calculated beam combination efficiency of the doughnut-shaped OV beams when the position of the MCF drifts in the x-direction resulting in a periodic occurrence of the doughnut-shaped beam which is also shifted in the x-direction but with a reduced combination efficiency. More precise adjustment of the position of the MLA will be required in our future work.

As mentioned in our response to Comment #2, the higher combining efficiency of the higher-order modes can be attributed to the greater similarity of the intensity distribution of the combined beam to the input beamlet arrays. Our numerical calculations show that the theoretical combination efficiency is ~46% for the Gaussian beam and ~60% for the doughnut-shaped OV and CV beams and detailed information is presented in Supplementary note 1.

Modification on Page 9, Paragraph 1:

“The amplified multiple beams were then combined with a MLA and a focusing lens and the combined beam was measured to be ~230 mW after a PH in **Error! Reference source not found.**(c), corresponding to a combination efficiency of ~44 % which is ~16% lower than the theoretical value of

~60%. The difference is probably due to the optical aberrations in the beam combination system induced by slight misalignment of the composite optical elements. For instance any deviation of the beam from the optical axis and inappropriately collimated beamlets can drastically reduce the combination efficiency (see more details in Supplementary note 4). The modal purity was measured to be ~93 % according to the vector mode decomposition method described in [47], indicating that the relative phase and polarization states of each core are well controlled.”

Figure R 3 (a) The calculated beam combination efficiency of the doughnut-shaped OV beam with a defocus on the MLA. (b) The calculated beam combination efficiency of the doughnut-shaped OV beam laterally shifted to different positions due to drift of the MCF position in the x-direction.

Comment 12: The authors claim that their system is essentially operating in open loop. If this is the case then, how sensitive is the situation to changes in temperature and/or the exact position of the fiber? How stable are the optimum solutions in time? When you go to the lab in the morning do you need to look for a new optimum solution each day or does the one from the previous day still work as intended?

Response: We can confirm that the system was operated in an open loop once the optimisation procedure had completed. The generated beam was quite stable and repeatable in the laboratory environment. As shown in the long-term stability measurement (Figure 7), which shows that the generated output beams were preserved for periods in excess of 1 hour (only small fluctuation of the merit value (~6% peak-to-valley) were observed over a 70 minute measurement period). It is to be appreciated that our adaptive coherent beam combination setup incorporates multiple optical components, including a microlens array, and maintaining a fixed position between the MCF and the external optics is critical to realising good long term stability. In our experiment, we observed some level of slow optical/mechanical drift over time and we found that we had to periodically re-adjust/re-align certain of the mounts to maintain good optical alignment every a few hours We believe that this is related to mechanical drift of the fiber and the MLA positioning stages (we used Thorlabs 3-axis microblock stages (MBT616D/M) in the experiment and have found some information on the long term stability of similar stages, e.g. 1.5 μm drift over 16 hours reported for the Newport ULTRAlign fiber stages [1]). To gain some confidence in this explanation we analysed the effect of positional drift by simulating the far-field intensity profiles of the shaped OV beam when the position of the MCF is moved along the x-direction. As shown in Figure R 4, a 1.5μm drift can result in severe beam distortion and can eventually decrease the beam combining efficiency.

When we went to the lab to run the system in the morning, the first thing we did was to adjust the position of the MCF and MLA to reduce the beam distortion due to the lateral drift as shown in Figure R 4. We then needed to run a new optimisation to ensure that the generated output beam had a high polarization extinction ratio and a high modal purity. (If we just used the solution from the previous day we experienced a slight degradation in polarization extinction ratio and modal purity).

We didn't directly investigate the temperature sensitivity of the system however we found that the shape of the output beam profile (or modal quality) is slightly dependent on the pump power level.

Figure R 4 The calculated far-field intensity profiles vs fibre positions in the x-direction.

Some examples are shown in Figure R 5. This might be related to thermal drift of mechanical/optical components, heat build-up inside the doped fibre, thermal distortion of the wavefront or depolarization of the combined beam due to gain variations. This is certainly an interesting topic for further investigation but beyond the scope of the current paper.

Figure R 5 The evolution of output beam profile with respect to the pump power.

Comment 13: The authors mention that conventional CBCs always require an active stabilization and control loop. This is inaccurate. If you only want to have the temporal stability shown in Fig.7, then you do not need any kind of feedback loop with CBC either. However, if you want to improve on that, then you would need a feedback loop both in conventional CBC and in the system presented by the authors. In essence, I do not think that their system is intrinsically more stable than a conventional CBC. Please justify or remove this sentence.

Response: We agree with the reviewer’s comment and the sentence is removed and revised accordingly.

Modification on Page 10, Paragraph 1:

“A small power fluctuation (~6% peak-to-valley) was observed with an average value of 0.979 and a root mean square variance of 0.007 over ~70 minutes, verifying stable polarization and phase locking across the six cores in our MCF amplifier. This implies very relaxed bandwidth requirements for CBC systems based on MCF amplifiers should active feedback stabilisation be deemed necessary in a given application.”

Comment 14: Provided that all these points are addressed I would have enough information to judge whether the results are good enough for Nature Communications (and in principle they certainly look like they are).

Response: We truly appreciate all the valuable comments and constructive suggestions from the reviewer. We have adopted all of their suggestions in our revised manuscript and feel that they these have improved our paper substantially. We hope that the reviewer is happy with our modifications and that they feel the paper is now suitable for publication in Nature Communications.

Reference:

[1] <https://www.newport.com/n/ultralign-precision-fiber-optic-positioning-system>

REVIEWERS' COMMENTS:

Reviewer #1 (Remarks to the Author):

My previous assessment was already favorable to accepting this paper for publication in Nature Communications. I now could read the other reviewer's report and the authors' answers and revisions and believe that they answered to all comments in a satisfactory way. The revised manuscript is even stronger than the first version and hence I confirm my recommendation of accepting the paper, as I think this work satisfies the criteria of importance and interest needed to be published in Nature Communications.

Reviewer #2 (Remarks to the Author):

I am happy with the changes made to the manuscript and I recommend the publication of the paper.

Best regards,